# Lipoteichoic Acid and Lipopolysaccharides Are Affected by p38 and Inflammatory Markers and Modulate Their Promoting and Inhibitory Effects on Osteogenic Differentiation

**DOI:** 10.3390/ijms232012633

**Published:** 2022-10-20

**Authors:** Kiyohide Ishihata, Chang-Hwan Seong, Toshiro Kibe, Kenta Nakazono, Fredy Mardiyantoro, Ryohei Tada, Masahiro Nishimura, Tetsuya Matsuguchi, Norifumi Nakamura

**Affiliations:** 1Department of Oral and Maxillofacial Surgery, Field of Maxillofacial Rehabilitation, Kagoshima University Graduate School of Medical and Dental Sciences, Kagoshima 890-8544, Japan; 2Department of Oral Biochemistry, Kagoshima University Graduate School of Medical and Dental Sciences, Kagoshima 890-8544, Japan; 3Department of Oral and Maxillofacial Prosthodontics, Kagoshima University Graduate School of Medical and Dental Science, Kagoshima 890-8544, Japan

**Keywords:** lipoteichoic acid, lipopolysaccharide, osteogenic differentiation, dental infection

## Abstract

Lipoteichoic acid (LTA) and lipopolysaccharide (LPS) are cell wall components of Gram-positive and Gram-negative bacteria, respectively. Notably, oral microflora consists of a variety of bacterial species, and osteomyelitis of the jaw caused by dental infection presents with symptoms of bone resorption and osteosclerosis. However, the effects of LTA and LPS on osteogenic differentiation have not yet been clarified. We examined the effects of LTA and LPS on osteoblasts and found that LTA alone promoted alizarin red staining at low concentrations and inhibited it at high concentrations. Additionally, gene expression of osteogenic markers (ALP, OCN, and OPG) were enhanced at low concentrations of LTA. High concentrations of LPS suppressed calcification potential, and the addition of low concentrations of LTA inhibited calcification suppression, restoring the gene expression levels of suppressed bone differentiation markers (ALP, BSP, and OCN). Moreover, the suppression of p38, a signaling pathway associated with bone differentiation, had opposing effects on gene-level expression of tumor necrosis factor-α (TNF-α) and interleukin-6 (IL-6), suggesting that mixed LTA and LPS infections have opposite effects on bone differentiation through concentration gradients, involving inflammatory markers (TNF-α and IL-6) and the p38 pathway.

## 1. Introduction

Dental infection is one of the most common inflammatory diseases in adults, which is caused by multiple species of bacteria, including Gram-positive and Gram-negative bacteria [1,2]. Furthermore, neglecting the management of a single tooth can have repercussions that extend to the jawbone. The clinical appearance of osteomyelitis of the jaw is often characterized by a special combination of bone resorption and osteosclerosis [3,4]. Management strategies for osteomyelitis of the jaw should consider both prevention and treatment of this condition [5]. Although there have been case reports suggesting that aggressive surgical intervention is counterproductive in the management of osteomyelitis of the jaw, more recently, there have been increasing reports supporting surgical intervention such as jaw reduction and reconstruction by vascular anastomosis [6]. However, the mechanism of development of osteomyelitis lesions in the jaw is still poorly understood.

Lipoteichoic acid (LTA) is an amphiphile located at the interface of the cytoplasmic membrane and cell wall of pathogenic and non-pathogenic Gram-positive bacteria and is released during growth [7,8]. The general perception of LTA is that it is strongly immunogenic and known to be a major virulence factor causing dental infections [9]. There is no consensus on the relationship between LTA and bone differentiation, with some reports suggesting that LTA effectively suppresses osteoclasts for promoting osteogenesis, suppresses osteoblasts, and has no effect on the osteogenic differentiation of dental pulp stem cells [8,9,10,11]. On the other hand, lipopolysaccharide (LPS), an extremely bioactive molecule generated by Gram-negative bacteria, functions prominently to increase the expression of osteoclastic cytokines such as interleukin-6 (IL-6), tumor necrosis factor-α (TNF-α), and receptor activator of nuclear factor-kappa B ligand [12,13,14]. Therefore, LPS plays an essential role in the process of inflammation-induced bone resorption and bone loss [15]. 

LPS induces immune responses, such as the proliferation of human immune cells and the release of inflammatory cytokines, and LTA is considered a Gram-positive strain of LPS because it has the same pathophysiological characteristics as LPS [16]. Therefore, these two bacterial toxins are expected to affect bone differentiation-related signals through the expression of inflammatory cytokines. For example, it has been suggested that LPS stimulates osteoblasts by activating mitogen-activated protein kinase (MAPK) signaling and nuclear factor-kappa B (NF-κB) signaling, leading to the production of various cytokines and contributing to systemic and local inflammatory responses [17]. Moreover, p38-MAPK modulates the transcriptional activation capacity of several important transcription factors in chondrocytes, osteoblasts, and osteoclasts, affecting their differentiation and function. p38 is related to the differentiation and proliferation of bone progenitors. p38-mediated phosphorylation promotes progression in osteogenesis by enhancing the activity and expression of osteoblast-specific transcription factors genes [18]. 

Although there are various reports on the effects of LTA and LPS on bone differentiation, most of them are based on LTA or LPS alone; there are few reports on the comparative study of both simultaneously [16,19]. We hypothesize that certain inflammatory signals are synergistically involved with osteoinductive signals in the pathogenesis of osteomyelitis of the jaw, which spills over from a mixed infection and induces bone formation or resorption by promoting or inhibiting osteoblast differentiation. To test this, we conducted experiments in which osteoblasts were seeded with LTA or LPS alone and in combination, using relatively high and low concentrations. We aimed to determine whether and how the environment created by the bacterial toxin affects the osteogenic potential of the osteoblasts.

## 2. Results

### 2.1. Effect of LTA on the Viability of Mouse Embryonic Osteoblast Precursor Cells

Considering that cell viability directly affects the response to exogenous stimulators, we first observed the morphology and viability of the mouse embryonic osteoblast precursor cell line MC3T3-E1. As shown in Figure 1, there were barely any changes in cell viability after 3 and 6 days when 10 μg/mL of LTA or less was added into the cell culture medium. However, the cell viability significantly decreased (*p* < 0.05) after 3 days with the application of 100 μg/mL of LTA.

### 2.2. Effects of LTA on the Formation of Mineralized Nodules

We examined whether LTA modulates osteogenic differentiation in MC3T3-E1 cells. To investigate the effect of LTA on MC3T3-E1 mineralization, MC3T3-E1 cells were cultured in an osteogenic induction medium supplemented with LTA (10 ng/mL and 5 μg/mL). Treatment with a low concentration of LTA promoted MC3T3-E1 mineralization at 3 and 14 days (Figure 2A,B). On day 3, treatment with 10 ng/mL of LTA significantly increased the ALP activity of MC3T3-E1 cells compared with the untreated group. On the other hand, stimulation with a high concentration (5 μg/mL) of LTA significantly suppressed ALP activity compared to the untreated condition on days 3 and 14 (Figure 2C,D).

### 2.3. Effects of LTA on the Expression of Osteogenic-Related Factors

We investigated the changes in mRNA levels of four bone-forming related factors (Sp7, ALP, OCN, and OPG) in different concentration groups. In the non-osteogenic medium with a low concentration of LTA, Sp7 and ALP were significantly upregulated on day 3 and then downregulated thereafter. OCN and OPG remained significantly upregulated until day 9 (Figure 3A). In the non-osteogenic medium with a high concentration of LTA, the expression of Sp7 and ALP increased on day 9, and the increase was about 2-fold compared to controls. OCN and OPG showed no change in expression (Figure 3B). Furthermore, we investigated the changes in the expression of bone-forming-related factors by exposure to different concentrations of LTA in an osteogenic medium. For this condition, low concentrations of LTA did not suppress the expression of Sp7, ALP, OCN, or OPG compared with controls; however, under high LTA concentrations, Sp7 expression was significantly suppressed on days 6 and 9, and ALP expression was suppressed on days 3 and 9. In addition, the expression of OCN and OPG continued to be significantly suppressed after day 3 (Figure 3C).

### 2.4. Relationship between Low-Concentration LTA and High-Concentration LPS on Bone Differentiation, Inflammatory Markers, and p38

Previous studies have shown that, in a non-osteogenic medium, low concentrations of LTA enhance bone-forming-related factors over time, and in an osteogenic medium, high concentrations of LTA markedly suppress bone-forming-related factors, whereas low concentrations of LTA have little effect. Next, to investigate the effect of mixed infection on bone differentiation, we examined the effects of LPS alone and in combination with low or high concentrations of LTA on osteoblast calcification and bone-forming-related factors. LPS alone and high concentrations of LTA and LPS inhibited osteoblast calcification, whereas low concentrations of LTA restored the inhibition of LPS-induced calcification (Figure 4A). The effect on the expression of bone differentiation markers, such as ALP, BSP, and OCN, was similar to that of calcification capacity, and LPS alone and high concentrations of LTA and LPS suppressed the expression of bone-forming-related factors, while low concentrations of LTA restored the suppression by LPS (Figure 4B). It is known that bacterial toxins induce strong expression of inflammatory markers such as TNF-α and IL-6, and thus, we also examined the expression of these markers. The expression of inflammatory markers was enhanced by LPS alone and by high concentrations of LTA and LPS, while low concentrations of LTA suppressed the expression of inflammatory markers induced by LPS, which was the opposite of the expression of bone-forming-related factors. Finally, the inhibition of p38 resulted in a further enhancement of TNF-α expression, but inhibited the suppression of IL-6 expression caused by low concentrations of LTA (Figure 4C).

## 3. Discussion

In the present study, we have demonstrated that low-concentration LTA promoted or maintained osteogenic differentiation, LPS inhibited osteogenic differentiation, and osteogenic differentiation inhibited by LPS was restored by low-concentration LTA. Low concentrations of LTA significantly enhanced the mineralization of osteoblasts during short-term observation, whereas high concentrations of LTA significantly inhibited it. In the non-osteogenic medium, stimulation with low concentrations of LTA significantly increased the expression of osteogenesis-related markers. In the osteogenic medium, the expression of osteogenesis-related markers was significantly suppressed by high concentrations of LTA, but was maintained after stimulation with low concentrations of LTA. LPS suppressed the calcification capacity of osteoblasts and the expression of bone differentiation-related markers and enhanced the expression of inflammatory markers, but both effects were reversed by the addition of low concentrations of LTA. Inhibition of p38 resulted in the recovery of suppressed IL-6 expression, suggesting that the osteosclerotic pattern in combined infections contributes in part to the association between IL-6 and p38 signaling. These results support the pathogenesis of osteosclerosis and bone resorption presented by osteomyelitis of the jaw, which spills over from dental infection, a typical example of a mixed infection. 

Hu et al. reported that LTA directly enhanced indicators of osteogenic factor-induced MC3T3-E1 cell differentiation, including alkaline phosphatase activity, calcium deposition, and osteopontin expression, and inhibited osteoclast activation by the receptor activator of nuclear factor-kappa B; therefore, LTA exhibits promising bone-regeneration effects [11]. LTA also promotes the expression of calcification potential and bone differentiation-related markers in a concentration-dependent manner in mesenchymal stem cells [20]. On the other hand, Yin et al. reported that LTA inhibits osteogenesis differentiation because LTA stimulation suppresses the formation of mineralized osteoblasts and the expression of bone differentiation-related markers [9]. Herein, we demonstrated that LTA alone at 10 ng/mL significantly enhanced the mineralization of MC3T3-E1 cells on day 3, while 5 μg/mL significantly inhibited it. In addition, although this was a relatively short-term study (up to day 9), early expression of the ALP gene and time-dependent enhancement of the OCN and OPG genes were observed in the non-osteogenic medium with 10 ng/mL of LTA, whereas 5 μg/mL of LTA had no significant effect on the expression of bone differentiation-related markers. On the other hand, 10 ng/mL of LTA had no obvious effect on accelerated bone differentiation in the osteogenic medium, while 5 μg/mL of LTA significantly inhibited the expression of the OCN gene in particular. A concentration of 10 ng/mL was used in the report that LTA promotes bone formation, and a concentration of 5 μg/mL was used in the report that LTA inhibits bone formation; in this study, experiments were conducted using these concentrations as references, and results were obtained with a consistent pattern. The discrepancy between our results and previous results may be due to the different cells and differences in the surrounding environment of the cells. In addition, the relevance of the concentration settings in the present study to the actual concentrations in infected foci needs to be discussed. However, there are many inhibitors of LTA and LPS in blood, and LTA and LPS themselves exist in various forms; problems such as discrepancies between blood levels and clinical symptoms have been pointed out, and there is currently no in vivo measurement method that can serve as a gold standard [21]. Therefore, verification of the involvement of LTA and LPS in vivo, including animal experiments, is required.

LPS has long been recognized as a potent inducer of the development of osteolytic bone loss and is capable of activating inflammatory cells, promoting secretion of proinflammatory cytokines, and inducing osteoclast precursor infusion and osteoclastic bone erosion [22,23]. Regarding the bone resorption mechanism of LPS, LPS may be involved directly in inflammatory bone loss and indirectly through the production of LPS-induced host factors such as IL-1 and TNF-α [22]. In the present study, LPS treatment significantly induced the expression of TNF-α and IL-6 and suppressed the expression of osteogenesis-related markers, such as ALP, BSP, and OCN, thereby inhibiting the mineralization of osteoblasts. The results suggest that the expression of inflammatory markers and osteogenesis show opposing responses. In previous reports, neither LPS nor LTA had a significant effect on ALP activity or calcium deposition during short-term toxin stimulation at any concentration, and after prolonged LPS stimulation at higher doses, ALP activity and calcium levels were shown to be significantly higher than those of untreated controls [16]. Extending these observations, the current study showed that, after LPS treatment, low-dose LTA restored osteoblast mineralization and expression of osteogenesis-related markers that were suppressed by LPS. We also established that high-dose LTA had no effect. Furthermore, treatment with a p38 inhibitor abolished the downregulation of IL-6 expression induced by low-dose LTA, while TNF-α expression was enhanced during stimulation with LPS alone and with LPS and high-dose LTA. These data suggest that p38-mediated osteogenic differentiation of osteoblasts is mainly induced by IL-6. Previous studies have shown that inflammatory responses induce calcification by activating p38 and that IL-6, in collaboration with IL-11, is responsible for transcriptional activation of human osteoblasts through the p38 system [17,24,25,26]. According to the present study, the pathogenesis of osteomyelitis of the jaw—which is known to be induced by a mixed infection, where bone resorption and osteosclerotic lesions are combined—is mediated at least in part by the activation of IL-6 and p38 signaling. Given the correlation of bone homeostasis that is maintained by a balance between osteoblasts and osteoclasts, inflammatory factors and osteogenic signaling may also play a role in osteoclast differentiation and activity. Although we believe that we have found some new findings in this study, there are still some issues that need to be verified. The present study did not explore the in vivo mechanisms, only a small portion of the signaling pathway was examined, and the protein level was not examined. More studies are necessary to further elucidate the pathogenesis of osteomyelitis of the jaw, which presents a complex pathologic picture. 

In conclusion, bone resorption induced by infection is elicited by enhanced expression of inflammatory cytokines and suppression of bone-related markers, and low-dose LTA restores LPS-induced bone resorption by inducing osteogenic differentiation of osteoblast through IL-6 and p38 signaling. In this study, we attempted in vitro experiments in which LTA and LPS were assumed to be the cause of mixed infections. As a result, we obtained one new trend in the pathogenesis of mixed infections, but we recognize that there is still a long way to go to elucidate the truth about osteomyelitis of the jaw, a classical disease of mixed infections.

## 4. Materials and Methods

### 4.1. Cell Culture and Osteogenic Differentiation

MC3T3-E1 cells were obtained from RIKEN Cell Bank (Tsukuba, Japan) and maintained in Eagle’s a-minimal essential medium (Sigma-Aldrich, Inc., St. Louis, MO, USA) containing 10% fetal bovine serum (FBS), 10 mM HEPES (pH 7.2–7.5), 100 units/mL penicillin, and 100 μg/mL streptomycin. To induce osteogenic differentiation, MC3T3-E1 cells were cultured in a growth medium (α-MEM supplemented with 10% FBS), and then cultured in an osteogenic induction medium (α-MEM supplemented with 10% FBS, 5 mM β-glycerophosphate, and 840 μM L-ascorbic acid-2-phosphate).

### 4.2. Antibodies and Reagents

The concentration of LTA was set at 10 ng/mL for low concentration, 5 μg/mL for high concentration, and 1 μg/mL for LPS. SB203080, a p38-specific inhibitor, was purchased from Funakoshi (Tokyo, Japan).

### 4.3. MTS Assay

For preparation, 5 × 103 MC3T3-E1 cells were inoculated into a 96-well plate and cultured for 24 h. Next, after discarding the culture medium, 100 μL of fresh DMEM medium containing 10% FBS and the various concentrations of LTA were added. The cells were treated with different concentrations (0.01, 0.1, 1, 10, and 100 μg/mL) of LTA for 3 and 6 days. Then, the MTS assay proceeded according to the manufacturer’s instructions. After pipetting 10 μL of CellTiter 96 AQ One Solution Reagent into each well of the assay plate, cells were incubated at 37 °C for 1 h. The absorbance was measured at 490 nm using a 96-well plate reader.

### 4.4. Alizarin Red Staining

Mineralization of MC3T3-E1 was evaluated by alizarin red staining. Cells were washed with Ca2+-free phosphate-buffered saline (PBS) three times and fixed in 10% formaldehyde/PBS for 20 min at 4 °C. After three washes with distilled water, the cells were stained in 1% alizarin red S solution for 5 min to visualize matrix calcium deposition. The excess staining was removed by several washes with distilled water, and the stained matrix was photographed.

### 4.5. Real-Time Reverse Transcription PCR Analysis

Total RNA was isolated from cells using Isogen II (Nippon Gene Co., Ltd., Tokyo, Japan) from MC3T3-E1 cells cultured in osteogenic or non-osteogenic induction medium supplemented with or without LTA (10 ng/mL or 5 μg/mL) and LPS (1 μg/mL) for various times. Reverse transcription was performed using the ReverTra Ace kit (Toyobo, Tokyo, Japan) according to the manufacturer’s instructions. Real-time PCR was conducted using a CFX ConnectTM system (Bio-Rad, Hercules, CA, USA). Briefly, the cDNA synthesized from 0.05 μg of total RNA was amplified in a volume of 20 μL with 0.11 x SYBR Green I (CAMBREX, Rockland, ME, USA), 0.2 mM/each of dNTPs, 0.5 μM/each of a pair of primers, and 0.5 unit Dream Taq Hot Start DNA polymerase (Thermo Fisher Scientific Inc., Waltham, MA, USA) under the following conditions: 95 °C for 5 min, followed by 55 PCR cycles at 95 °C for 30 s, 60 °C for 20 s, and 72 °C for 40 s. Fluorescent signals were measured in real time, and then each sample was quantified according to the manufacturer’s instructions. The primer sequences used in this study are listed in Table 1. To normalize the differences in the amount of total RNA added to each reaction, ribosomal protein L13a (Rpl13a) was used as the endogenous control. An arbitrary unit was determined by dividing the concentration of each PCR product by the concentration of the Rpl13a PCR product. Each real-time PCR analysis was performed in triplicate and repeated at least three times to confirm consistent results.

### 4.6. Statistical Analysis

The data are presented as means ± standard deviation (S.D.). Comparisons between two groups were performed using Student’s *t*-test. Statistical analysis for multiple comparisons among the groups was performed using one-way ANOVA, and *p* < 0.05 was considered to indicate a statistically significant difference.

## Figures and Tables

**Figure 1 ijms-23-12633-f001:**
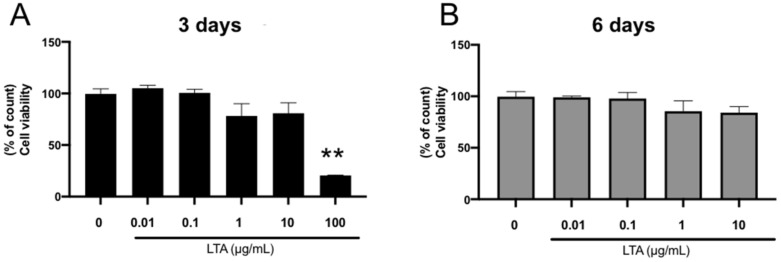
High concentrations of LTA inhibited MC3T3-E1 cell proliferation. MC3T3-E1 cells were seeded in 96-well plates at 2 × 10³ cells per well with a 100 μL medium. (**A**) MC3T3-E1 cells were treated with different concentrations (0.01, 0.1, 1, 10, and 100 μg/mL) of LTA for 3 days. The MTS assay proceeded according to the manufacturer’s instructions. (**B**) MC3T3-E1 cells were treated with different concentrations (0.01, 0.1, 1, and 10 μg/mL) of LTA for 6 days. The MTS assay proceeded according to the manufacturer’s instructions. After pipetting 10 μL of CellTiter 96 AQ One Solution Reagent into each well of the assay plate, cells were incubated at 37 °C for 1 h. The absorbance was measured at 490 nm using a 96-well plate reader. Cell survival was not affected at concentrations up to 10 μg/mL at either 3 or 6 days of incubation. Results are presented as means ± S.D. ** *p* < 0.01.

**Figure 2 ijms-23-12633-f002:**
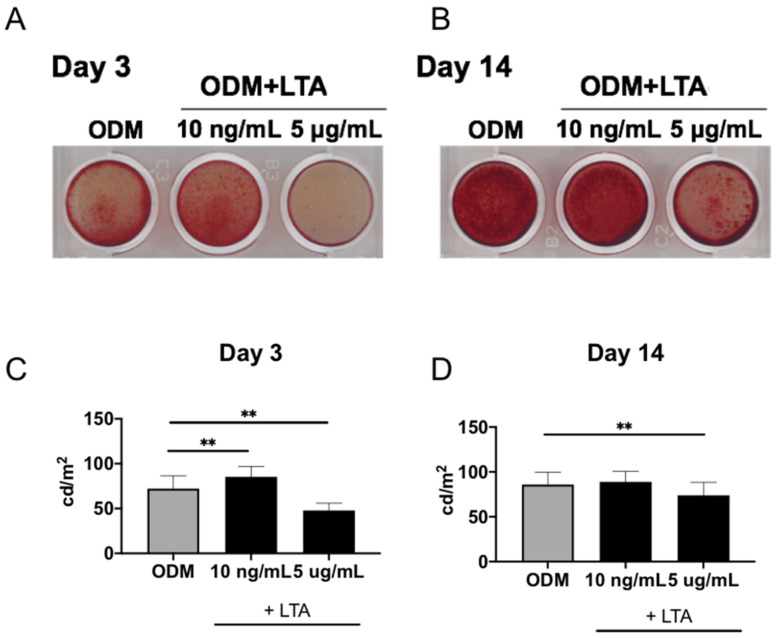
Effects of LTA on MC3T3-E1 osteogenic differentiation. (**A**,**B**) MC3T3-E1 cells were cultured in an osteogenic induction medium supplemented with or without LTA (10 ng/mL or 5 μg/mL) for the indicated times. MC3T3-E1 mineralization was investigated by alizarin red staining. (**C**,**D**) MC3T3-E1 cells were cultured in an osteogenic induction medium supplemented with or without LTA (10 ng/mL or 5 μg/mL) for 3 or 14 days, and then MC3T3-E1 mineralization was measured (n = 3). On day 3, after addition of LTA to the osteogenic induction medium, mineralization was significantly enhanced at 10 ng/mL and significantly inhibited at 5 μg/mL; on day 14, mineralization was significantly inhibited at 5 μg/mL, while no change was observed at 10 ng/mL. Results are presented as means ± S.D. ** *p* < 0.01.

**Figure 3 ijms-23-12633-f003:**
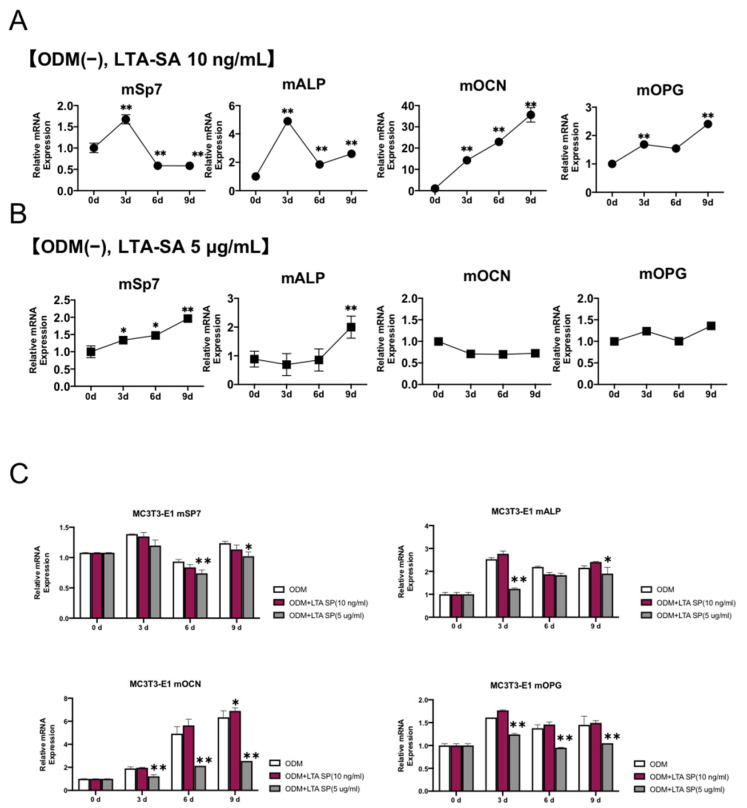
Effects of LTA on MC3T3-E1 osteogenesis-related markers. (**A**,**B**) Expression of osteogenic marker genes in MC3T3-E1 cells cultured in non-osteogenic induction medium supplemented with LTA (10 ng/mL or 5 μg/mL) for the indicated times. At 10 ng/mL, Sp7 and ALP showed a significant temporary increase in expression, and OCN and OPG showed a time-dependent increase in expression until day 9. A value of 5 μg/mL showed an increase in Sp7 and ALP on day 9, but no change in OCN and OPG expression. (**C**) Expression of osteogenic marker genes in MC3T3-E1 cultured in osteogenic induction medium supplemented with LTA (10 ng/mL or 5 μg/mL) for the indicated times. In the osteogenic induction medium, the addition of LTA did not enhance the expression of osteogenesis-related markers. At 5 μg/mL, the expression of SP7 on days 6 and 9, ALP on days 3 and 9, and OCN and OPG on all measurement days were suppressed. ALP—alkaline phosphatase; OCN—osteocalcin; OPG—osteoprotegerin. Results are presented as means ± S.D. * *p* < 0.05, ** *p* < 0.01.

**Figure 4 ijms-23-12633-f004:**
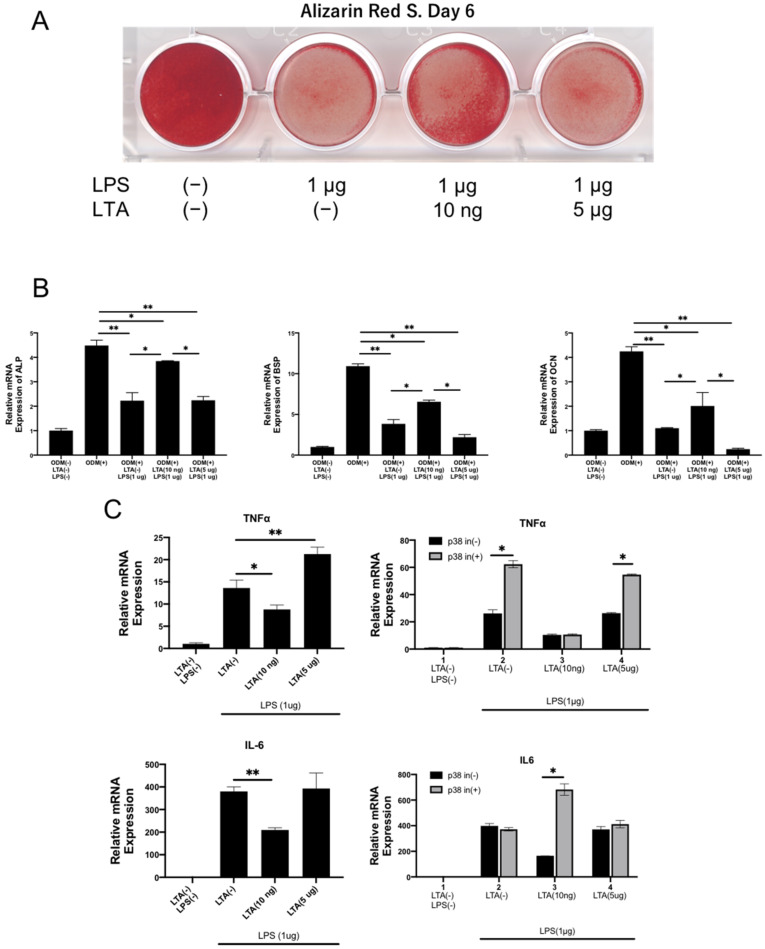
Involvement of TNF-α, IL-6, and p38 signaling in LPS- and LTA-stimulated osteogenic differentiation of MC3T3-E1. (**A**) Effects of LPS and LTA on osteogenic differentiation of MC3T3-E1 cells. MC3T3-E1 cells were cultured in an osteogenic induction medium with LPS only (1 μg/mL) or combined with LTA (10 ng/mL or 5 μg/mL) for 6 days. MC3T3-E1 mineralization was evaluated by alizarin red staining. Addition of LPS alone inhibited mineralization, and supplementation with 10 ng/mL of LTA restored mineralization inhibited by LPS. A value of 5 μg/mL of LTA did not lift mineralization inhibition. (**B**) Expression of osteogenic marker genes in MC3T3-E1 cells cultured in an osteogenic induction medium supplemented with only LPS (1 μg/mL) or combined with LTA (10 ng/mL or 5 μg/mL) for 6 days (n = 3). LPS alone significantly suppressed the expression of ALP, BSP, and OCN, and the addition of 10 ng/mL of LTA significantly restored the expression of osteogenic marker genes; the addition of 5 μg/mL of LTA further suppressed the expression of osteogenic marker genes. ALP—alkaline phosphatase; BSP—bone sialoprotein; OPG—osteoprotegerin. (**C**) Expression of inflammatory marker genes in MC3T3-E1 cultured in an osteogenic induction medium supplemented with only LPS (1 μg/mL) or combined with LTA (10 ng/mL or 5 μg/mL) for 6 days (n = 3). Effects of SB203080, a p38-specific inhibitor, induced osteogenic differentiation of MC3T3-E1 cells. The expression of inflammatory markers TNF-α and IL-6 was enhanced by LPS alone and by the addition of 5 μg/mL of LTA, and was suppressed by the addition of 10 ng/mL of LTA, in symmetry with the results of mineralization experiments. p38 inhibition restored IL 6 expression, which was suppressed by 10 ng/mL LTA, and IL6 expression was enhanced by p38 inhibition. Results are presented as mean ± S.D. * *p* < 0.05, ** *p* < 0.01.

**Table 1 ijms-23-12633-t001:** Primer sequences used in this study.

Gene	Primer Sequence [5′-3′]		Product Size
*Rpl13a*	Forward	GCTTACCTGGGGCGTCTG	149 bp
	Reverse	ACATTCTTTTCTGCCTGTTTCC	
*TNF-α*	Forward	TCCCCAAAGGGATGAGAAGTT	345 bp
	Reverse	GAGGAGGTTGACTTTCTCCTGG	
*IL-6*	Forward	CAACGATGATGCACTTGCAGA	142 bp
	Reverse	CTCCAGGTAGCTATGGTACTCCAGA	
*Sp7*	Forward	TATGCTCCGACCTCCTCAAC	120 bp
	Reverse	AATAGGATTGGGAAGCAGAAAG	
*Alp*	Forward	GGCTTCTTCTTGCTGGTGGAA	97 bp
	Reverse	CCTGGTCCATCTCCACTGCT	
*Bsp*	Forward	AGGGAACTGACCAGTGTTGG	124 bp
	Reverse	ACTCAACGGTGCTGCTTTTT	
*Ocn*	Forward	CTCACAGATGCCAAGCCCA	98 bp
	Reverse	CCAAGGTAGCGCCGGAGTCT	

Rpl13a: housekeeping gene, TNF-α: tumor necrosis factor-α, IL-6: Interleukin-6, Sp7: Transcription factor Sp7, Alp: alkaline phosphatase, Bsp: Bone sialoprotein, Ocn: osteocalcin.

## Data Availability

Data are available on request due to privacy or other restrictions.

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
