# Peer review of "Lipoteichoic Acid and Lipopolysaccharides Are Affected by p38 and Inflammatory Markers and Modulate Their Promoting and Inhibitory Effects on Osteogenic Differentiation"

_ijms, 2022, doi:10.3390/ijms232012633_

Round 1
Reviewer 1 Report
The research was conducted based on the pathophysiology of jaw osteomyelitis with both bone resorption and osteosclerosis, which is an interesting research direction in clinical practice. The authors cultured osteoblasts in different concentrations of LTA or LPS alone and in combination to explore the mechanisms of jaw bone mixed infection. The idea is new, but several questions are still to be solved before this manuscript can be reconsidered.
1) What is the reason of choosing 10ng/L and 5ug/L of LTA as "low" and "high" concentration in the study design?
2) I am curious that, under common infection condition, what concentrations of LTA and LPS would be expected at the site of infection. And in which medium should these concentrations be measured? (eg, blood, tissue fluid...)
3) In Discussion section, the authors claimed that "the osteosclerotic pattern contributes primarily to the association between IL-6 and p38 signaling", which I found difficult to understand. I believe more experiments are needed to prove this conclusion.
4) Animal experiments are expeccted to provide stronger proof.
5) Writing of th manuscript has to be improved. For example, figure legends should present briefly about the experimental results instead of adding detailed description about methods.
Author Response
Response to Reviewer 1.
We wish to express our appreciation to the Reviewer for his or her insightful comments, which have helped us significantly improve the paper.
General Comments
The research was conducted based on the pathophysiology of jaw osteomyelitis with both bone resorption and osteosclerosis, which is an interesting research direction in clinical practice. The authors cultured osteoblasts in different concentrations of LTA or LPS alone and in combination to explore the mechanisms of jaw bone mixed infection. The idea is new, but several questions are still to be solved before this manuscript can be reconsidered.
Response to general comments: We thank the Reviewer for this pertinent comment. The manuscript has been revised to address the points raised by the reviewers. In addition, the manuscript has been edited by a professional English editing service, and we would appreciate it if you would review it again.
Specific Comments
1) What is the reason of choosing 10ng/L and 5ug/L of LTA as "low" and "high" concentration in the study design?
Response 1) We thank the reviewer for this comment. To answer the reviewer's questions, we have added the following to the second paragraph of the discussion section. “ A concentration of 10 ng/mL was used in the report that LTA promotes bone formation, and a concentration of 5 μg/mL was used in the report that LTA inhibits bone formation, and in this study, experiments were conducted using these concentrations as references and results were obtained with a consistent pattern.”
2) I am curious that, under common infection condition, what concentrations of LTA and LPS would be expected at the site of infection. And in which medium should these concentrations be measured? (eg, blood, tissue fluid...)
Response 2) We thank the reviewer for this comment. As you pointed out, the experimental design should be designed with reference to the concentration of bacterial toxins in vivo. However, it is said that there are many inhibitors for extracting bacterial toxins in vivo, and it is difficult to measure their concentrations. We have added the following in the second paragraph of the Discussion section, giving the references cited. “In addition, the relevance of the concentration settings in the present study to the actual concentrations in infected foci needs to be discussed. However, there are many inhibitors of LTA and LPS in blood, and LTA and LPS themselves exist in various forms, and problems such as discrepancies between blood levels and clinical symptoms have been pointed out, and there is currently no in vivo measurement method that can serve as a gold standard(Valenza et al. 2009). Therefore, verification of the involvement of LTA and LPS in vivo, including animal experiments, is required.”
3) In Discussion section, the authors claimed that "the osteosclerotic pattern contributes primarily to the association between IL-6 and p38 signaling", which I found difficult to understand. I believe more experiments are needed to prove this conclusion.
Response 3) We thank the reviewer for this comment. We agree that additional information on more experiments as the reviewer suggested would be valuable. Regrettably, however, because of limitation of timeand as follow, we are unable to do the experimentation. We are aware that we need to examine the case of phosphorylation of signaling pathways, verification of protein level, verification of mineralization accordingly and verification of bone differentiation markers as future studies, but we would like to summarize the results of gene level experiments in this study. Therefore, first of all, the first paragraph in the Discussion section has been modified as follows, “ Inhibition of p38 resulted in the recovery of suppressed IL-6 expression, suggesting that the osteosclerotic pattern in combined infections contributes in part to the association between IL-6 and p38 signaling.” In addition, the third paragraph of the Discussion section has been added as follows, “Although we believe that we have found some new findings in this study, there are still some issues that need to be verified. The present study did not explore the in vivo mechanisms, only a small portion of the signaling pathway was examined, and the protein level was not examined.” I hope you will forgive me for the above responses.
4) Animal experiments are expeccted to provide stronger proof.
Response 4) We thank the reviewer for this comment. This is a duplicate of point 2, but I would like to respond as follows. We believe that in vivo experiments are necessary to verify the onset of osteomyelitis, as you have pointed out. However, it is said that there are many inhibitors for extracting bacterial toxins in vivo, and it is difficult to measure their concentrations. We have added the following in the second paragraph of the Discussion section, giving the references cited. “In addition, the relevance of the concentration settings in the present study to the actual concentrations in infected foci needs to be discussed. However, there are many inhibitors of LTA and LPS in blood, and LTA and LPS themselves exist in various forms, and problems such as discrepancies between blood levels and clinical symptoms have been pointed out, and there is currently no in vivo measurement method that can serve as a gold standard(Valenza et al. 2009). Therefore, verification of the involvement of LTA and LPS in vivo, including animal experiments, is required.” In addition, I added the following to the third paragraph of the discussion section, “Although we believe that we have found some new findings in this study, there are still some issues that need to be verified. The present study did not explore the in vivo mechanisms, only a small portion of the signaling pathway was examined, and the protein level was not examined.”
5) Writing of the manuscript has to be improved. For example, figure legends should present briefly about the experimental results instead of adding detailed description about methods.
Response 5) We thank the reviewer for this comment. In Figure legend, I added the following explanation of the experimental results.
Figure 1; Cell survival was not affected at concentrations up to 10 μg/ml at either 3 or 6 days of incubation.
Figure 2; On day 3 after addition of LTA to the osteogenic induction medium, mineralization was significantly enhanced at 10 ng/ml and significantly inhibited at 5 μg/ml; on day 14, mineralization was significantly inhibited at 5 μg/ml while no change was observed at 10 ng/ml.
Figure 3;
At 10 ng/ml, Sp7 and ALP showed a significant temporary increase in expression, and OCN and OPG showed a time-dependent increase in expression until day 9. 5 μg/ml showed an increase in Sp7 and ALP on day 9, but no change in OCN and OPG expression.,
In osteogenic induction medium, the addition of LTA did not enhance the expression of osteogenesis-related markers. At 5 μg/ml, the expression of SP7 on days 6 and 9, ALP on days 3 and 9, and OCN and OPG on all measurement days were suppressed.
Figure 4;
Addition of LPS alone inhibited mineralization, and supplementation with 10 ng/ml of LTA restored mineralization inhibited by LPS. 5 μg/ml of LTA did not lift mineralization inhibition.
LPS alone significantly suppressed the expression of ALP, BSP, and OCN, and the addition of 10 ng/ml of LTA significantly restored the expression of osteogenic marker genes; the addition of 5 μg/ml of LTA further suppressed the expression of osteogenic marker gene.
The expression of inflammatory markers TNF-α and IL-6 was enhanced by LPS alone and by the addition of 5 μg/ml of LTA, and was suppressed by the addition of 10 ng/ml of LTA, symmetrically with the results of mineralization experiments. p38 inhibition restored IL 6 expression, which was suppressed by 10 ng/ml LTA, was restored and IL6 expression was enhanced by p38 inhibition.

Reviewer 2 Report
The manuscript addresses a modern and multidisciplinary (dentistry, microbiology, molecular biology, maxillary surgery) research topic with an impact on pathogenesis and management of osteomyelitis of the jaw, characterized by symptoms of bone resorption and osteosclerosis and caused by dental infection. It was studied the effects of two bacterial toxins as lipoteichoic acid (LTA) and lipopolysaccharide (LPS), on osteogenic differentiation process. The originality of the research consists in the fact that osteoblasts were seeded with LTA or LPS alone and in combination, using relatively high and low concentrations, while other scientific reports are based on LTA or LPS alone and there are only few reports on the comparative study of both simultaneously.

Author Response
Response to Reviewer 2.
General Comments
The manuscript addresses a modern and multidisciplinary (dentistry, microbiology, molecular biology, maxillary surgery) research topic with an impact on pathogenesis and management of osteomyelitis of the jaw, characterized by symptoms of bone resorption and osteosclerosis and caused by dental infection. It was studied the effects of two bacterial toxins as lipoteichoic acid (LTA) and lipopolysaccharide (LPS), on osteogenic differentiation process. The originality of the research consists in the fact that osteoblasts were seeded with LTA or LPS alone and in combination, using relatively high and low concentrations, while other scientific reports are based on LTA or LPS alone and there are only few reports on the comparative study of both simultaneously.
Strengths of manuscrip
The manuscript corresponds to the stated purpose and objectives of the journal.
The title accurately reflects the content of the paper.
The abstract is structured as a short synopsis of the paper.
The introduction presents in detail the current level of knowledge on the approachedsubject and highlights why this research is important. This introduction is comprehensible to scientists outside the research field. The purpose of the work and its significance are defined as one hypothesis and
one objective. Also, the introduction includes 19 relevant references, out of which 9 are published between years 2017 - 2022s
The discussion present an interpretation of the results in perspective of the previous studies and the purpose and working hypotheses of the study. The findings and their implications are discussed in the broadest context possible.
The conclusions are presented in generally manner and highlight the implications of dental infection in bone resorption by enhanced expression of inflammatory cytokines and supression of bone-related makers. The conclusions mentioned that low-dose LTA restores LPS-induced bone resorption by inducing osteogenic differentiation of osteoblast through IL-6 and p38 signaling. They are interesting for the readership of the journal.
The manuscript contains a complex, but correctly designed and technically sound research method. The chapter “Materials and Methods” clearly presents a large number of research steps. The instruments, research equipment, software programs and reagents are described in sufficient details to allow another research team to reproduce the results. Also, the statistical analyses methods are well described.
The performed analyses are appropriate. The research results are presented at higher standards, including 4 figures. The way of reporting the results was elaborated taking into account the steps of the research method. The images are significant and suggestive for the research subject. The results are presented concisely and systematically. All figures are elaborated according to authors’ guideline. The references are in accordance with the studied topic. The manuscript contains 25
references, most representing studies published after the year 2015.
Weakness of the manuscript
None
Response to Reviewer 2.
We are very impressed with your careful review results. Thank you very much for your kind comments. We would like to continue our efforts for the development of medicine.
We express our gratitude.

Round 2
Reviewer 1 Report
The authors have revised the manuscript and have responded most suggestions mentioned. Nevertheless, no experiment has been added in the revision. Although some discussion considering my former concerns has been added, I am afraid the underlying mechanisms of jaw osteomyelitis is still insufficient in the present article.
Author Response
Response to Reviewer 1.
We wish to express our appreciation to the Reviewer for his or her insightful comments, which have helped us significantly improve the paper. We also thank the reviewer for taking time out of their busy schedules to re-read the edited manuscript carefully and for their appropriate comments.
General Comments
The authors have revised the manuscript and have responded most suggestions mentioned. Nevertheless, no experiment has been added in the revision. Although some discussion considering my former concerns has been added, I am afraid the underlying mechanisms of jaw osteomyelitis is still insufficient in the present article.
Response to general comments: We thank the Reviewer for this pertinent comment. Your point is correct and irrefutable. We are not convinced that we have elucidated the true nature of osteomyelitis of the jaw from the results of this study, and we are aware that a more thorough research plan is needed in the future. We have revised the conclusion column as follows, taking into consideration the intent of your suggestion to the maximum extent. “In this study, we attempted in vitro experiments in which LTA and LPS were assumed to be the cause of mixed infections. As a result, we obtained one new trend in the pathogenesis of mixed infections, but we recognize that there is still a long way to go to elucidate the truth about osteomyelitis of the jaw, a classical disease of mixed infections.” Would you permit publication in the journal such sentences. We would appreciate your reconsideration.

Round 3
Reviewer 1 Report
Dear authors,
Thanks for your response. I agree that this is a good point for future research and I look forward to see your further works on this topic.